# Occurrence of Alexithymia and Its Association with Sports Practice from a Sample of University Students: Results from a French Cross-Sectional Study

**DOI:** 10.3390/healthcare10050788

**Published:** 2022-04-23

**Authors:** Catarina Proença Lopes, Edem Allado, Aziz Essadek, Mathias Poussel, Audrey Henry, Eliane Albuisson, Aghilès Hamroun, Bruno Chenuel

**Affiliations:** 1Development, Adaptation and Disadvantage, Cardiorespiratory Regulations and Motor Control (EA 3450 DevAH), University of Lorraine, 54000 Nancy, France; e.allado@chru-nancy.fr (E.A.); m.poussel@chru-nancy.fr (M.P.); b.chenuel@chru-nancy.fr (B.C.); 2Center of Sports Medicine and Adapted Physical Activity, CHRU-Nancy, University of Lorraine, 54000 Nancy, France; 3INTERPSY (EA 4432), University of Lorraine, 54000 Nancy, France; aziz.essadek@univ-lorraine.fr; 4Cognition, Health and Society Laboratory, University of Reims Champagne-Ardenne, 51571 Reims, France; audrey.henry@univ-reims.fr; 5Institut Elie-Cartan de Lorraine, CNRS, Université de Lorraine, 54000 Nancy, France; e.albuisson@chru-nancy.fr; 6Department of Public Health, Epidemiology, Health Economics and Prevention, Regional and University Hospital Center of Lille, Lille University, 59000 Lille, France; aghiles.hamroun@chru-lille.fr

**Keywords:** alexithymia, athletic student, non-athletic student, emotion

## Abstract

Background: This study aimed to assess the prevalence of alexithymia in a sample of university students and to determine its association with specific sports practice characteristics (competition and training). Methods: In this cross-sectional study, anthropometric data and characteristics of sport practice were collected, as well as level of alexithymia (Toronto Alexithymia Scale, (TAS-20)). Results: The study included 253 French university students who completed a questionnaire specifying their regular sports practice and level of alexithymia (TAS-20). We found 76 subjects (30%) who had proven alexithymia and 92 (36.4%) who were borderline alexithymic. A significant positive relationship between alexithymia and the weekly amount of training practice was observed. It should be noted that students who engage in more than 5 h of physical activity are more prone to be borderline or alexithymic (respectively, 19.6 and 19.7% versus 7.1% for non-alexithymics; *p* = 0.03). Conclusion: With a 30% frequency, alexithymia is more prevalent in this context than in the general population. Furthermore, alexithymia and borderline alexithymia are most favorably associated with higher physical activity (over 5 h per week).

## 1. Introduction

Students are at risk of developing depression, anxiety, stress, suicidal tendencies [1,2] or other psychological disorders such as addictions. Indeed, students are susceptible to several types of addiction [3]. More than half of students commonly use drugs [4]. They are prone to using marijuana, alcohol and non-medical prescription drugs [5]. There are several reasons for this behavior, including leaving the family home, a changing lifestyle, the huge pressure to succeed, and post-graduation plans [1].

Numerous studies have evaluated anxiety and depressive disorders in the general population and have demonstrated a relationship with alexithymia [6,7,8,9,10,11]. Alexithymia refers to a deficit in the ability to recognize and express emotions, use of concrete speech and thoughts related to external events, and a paucity of notional thought processes [12]. It is linked to a number of psychosomatic illnesses and could represent vulnerability factors for the development of psychiatric disorders in adulthood [13], addictions such as eating disorders [14], and drug misuse [15]. Alexithymics are also at high risk for suicide [16]. Given the comorbidities of depression, anxiety or even addictions with alexithymia, whether students are alexithymic or not is an important question, and some recent works have sought to explore it. Alzahrani et al., [17] showed a strong presence of alexithymia within the student population, with 49% of medicine students being alexithymic (95% confidence interval [43.8–54.2])). The study by Lyvers et al. [18] demonstrated that, in a student population, lack of resilience is associated with more anxiety and problematic drinking due to higher levels of alexithymia. Mei et al., [19] observed that higher levels of alexithymia led to an increase in mobile phone addiction. Hošková-Mayerová and Mokrá [20] were interested in studying alexithymia in students of different disciplines. They demonstrated higher rates of alexithymia for life science students when compared with psychology students. Parker et al., [21] studied the impact of alexithymia on a major life transition (high school to university) and showed that the non-alexithymic group performed significantly better in their first year of study than the alexithymic group. 

The observation of a clear relationship between alexithymia and students is not surprising, as there are various stressors involved in experiencing a significant period of development, such as being at university [22], coupled with poor coping strategies, habitually used by individuals scoring high on measures of alexithymia [23].

It is known that sport is a protective factor of anxiety and depression as it can improve symptoms for these disorders [24]. It has been demonstrated that sports are widely associated with alexithymia [25], but in a negative way. In fact, researchers observed a link between alexithymia and overtraining [26,27,28,29], with addiction [30,31], and especially in high-risk sports [32,33,34,35,36,37]. However, for Barlow et al., [38] the predictive role of alexithymia remained significant when controlling for sensation seeking.

The research of Jodat and colleagues [39] on athletes and non-athlete students showed higher levels of alexithymia in the non-athlete cohort. The authors argued that physical activity can provide opportunities to divert the focus away from stressful conditions. It can also create a context for improving mental health, and emotional and physical well-being. Another study on students [40] demonstrated that alexithymia leads to exaggerated practice mediated by concerns over body image. This results in a positive self-image, leading to the perpetuation of sporting activity.

Meanwhile, Woodman et al. [41] observed that extreme endurance running provides an emotion regulation function for athletes with high alexithymia, which, paradoxically, leads to a decrease in alexithymia. 

We wanted to assess the occurrence of alexithymia in a student population and observe whether the practice of sport was particularly associated with alexithymia.

## 2. Materials and Methods

### 2.1. Participants

This cross-sectional study was performed from 1 February to 28 April 2017 outside of any influencing stressful factors (beginning of the academic year, exam period: early January and end of May). The participants in the study were undergraduates studying literature and human sciences at the University of Reims Champagne-Ardenne (France). All participants were consecutively recruited on a voluntary basis. The purpose of the study was explained orally and questionnaires were collected just after the explanation. Informed consent was obtained from all subjects. Inclusion criteria were minimum age 18 years and the ability to understand the French language. Exclusion criterion was the presence of a evolutional or chronic psychiatric disease, as reported by the subject. Only complete sets of data were analyzed (all questions in the questionnaire had to be completed). All participants received the same instructions from the same person (the principal investigator). Based on our recent work on a French population of sports-licensed young adults [37], with an expected 30% prevalence of alexithymia in the athletic group and 15% in the non-athletic group, the sample size was calculated at 242 (two-sided significance level of 5%, study power of 80% and same number of subjects in both groups).

### 2.2. Data Collection

For all subjects, we collected anthropometric data and the characteristics of sport practice, namely, type of sport, level of practice, and weekly amount of sport training (self-reported questionnaires).

Factors that might distort the true association and/or influence the interpretation of the results were carefully considered. We then took into account the following potential confounders: -Employment status (work while studying);-Drug treatment in progress;-If the person had suffered a cranial trauma, and if yes, date of the event; -Distance from home to university (the further home is, the more difficult it is to find time for physical activity);-Reasons for participating in a sports activity.

The standardized and validated Toronto Alexithymia Scale (TAS-20) questionnaire [42] was used to assess alexithymia and psychological factors in order to limit classification bias. Overall alexithymia score (high scores equate to high alexithymia) results from the combination of three subscales. A score greater than or equal to 56 is considered as ‘proven alexithymia’ and a score between 45 and 55 as a ‘borderline’ status. The three-factor structures are: difficulty identifying feelings (DIF) incorporating 7 items, difficulty describing feelings (DDF) with 5 items, and externally oriented thinking (EOT) with 8 items. Both the reliability and the validity of the TAS-20 have been amply demonstrated. Internal consistency was good for the entire TAS-20 scale (alpha = 0.79), the DIF (alpha = 0.77), and the DDF (alpha = 0.72) subscales, but the internal consistency of the EOT subscale was only moderate (alpha = 0.61).

### 2.3. Data Analysis

Categorical variables are presented as frequencies and percentages and quantitative variables as mean ± standard deviation (SD) or median (1st; 3rd quartile) according to their distribution. Comparative analyses were carried out using Student, Wilcoxon or ANOVA tests for continuous variables and chi-squared or Fisher’s exact test for categorical variables. The effect size calculation for the pairwise comparisons was carried out using Cramér’s V statistics. 

The α risk was established at 0.05. We used IBM™ SPSS Statistics Software v23 for data analysis.

### 2.4. Legal Obligation

All participants received clear and fair information to enable them to provide informed consent. The study is consistent with the ethical principles of the Helsinki declaration.

## 3. Results

In total, 275 responses were received, with 22 being excluded because of incomplete questionnaires. Thus, we analyzed 253 questionnaire responses. Participants were 20.6 (3.0) years on average and most were women (76.7%). There were 85 athletic students (students who practice a physical activity for at least 1 session per week), and the most common types of sports were endurance sports (28.2%). Among this sample, 85 students (33.6%) were identified as athletes, and there were more males than non-athletic students (31.8% versus 19.1%, respectively, *p* = 0.019) (Table 1).

In the overall population of 253 students, rates of borderline and proven alexithymia were 36.4% (*n* = 92) and 30.0% (*n* = 76), respectively. Alexithymics were slightly younger than borderline or non-alexithymic participants (19.9 (1.3), 20.7 (4.2) and 21.1 (2.4) years, respectively; *p* < 0.001), and tended to be more often women. Students engaged in more than 5 h of physical activity per week were more often borderline or alexithymic (19.6% and 19.7%, respectively versus 7.1% for non-alexithymics; *p* = 0.03), with a small size effect of this association (Cramer’s V = 0.145, 95 CI [0.096–0.234]). No significant difference was identified regarding level of practice (Table 2).

## 4. Discussion

We found alexithymia prevalence of 30% in our sample of French university students. A small but significant relationship was also observed between alexithymia and weekly duration of training practice, with alexithymia being more prevalent in subjects practicing sport for more than 5 h per week. However, no significant association was observed between alexithymia/borderline alexithymia and level of practice.

We observed that alexithymia prevalence was higher in the students in our study than in the general population (between 17% and 23%) [13]. This is probably because students more prone to anxiety and depression are more likely to feel unwell [43]. Indeed, an important association between depression and alexithymia has already been highlighted [44]. However, the causal relationship between the two psychological traits remains unknown. The student population is also more exposed to various sources of addiction, such as alcohol and drugs [45]. This may be one of the explanations for the high incidence of alexithymia. 

The significant relationship between alexithymia and sport has already been demonstrated in students engaged in sporting activities [28]. However, more broadly, few data exploring the association between alexithymia and sports, with highly heterogenous conclusions, were shown by our research team through a systematic review [25]. In fact, higher scores in alexithymia can be associated with being an athlete [46] as well as a non-sport students [39]. Based upon the common relationship between alexithymia and addiction, we could hypothesize that those students engaged in sport would be addicted to sport and therefore possibly suffer from bigorexia. Furthermore, we observed a significant relationship between the duration of weekly training and so-called borderline alexithymia and proven alexithymia. These results are in line with a study by Allegre et al. [28], which showed more pronounced alexithymic characteristics in professional swimmers who trained for 22 h per week compared to recreational swimmers practicing 6 h each week. However, our study is made up of both alexithymic and alexithymic borderline students, who mostly practice for more than 5 h in a sports-based leisure activity, suggesting that this association may not be confined to professional athletes alone. These results are also in line with previous research conducted by our team. Indeed, we recently showed an association between alexithymia and the duration of training [37]. Research on sport practitioners of different levels of activity has shown that alexithymic athletes were more involved in competition than recreational sports, and therefore engaged in more hours of training. Iacolino and colleagues [40] highlighted that excessive sport practice can lead to addiction because of the association of bodily misperception with an inability to recognize emotions. As stated previously, alexithymia leads to exaggerated practice, mediated by anxiety over body image, causing positive feelings that perpetuate sportive behavior. We hypothesize that, due to the variety of stressors at university, a coping mechanism could be involved. Alexithymia could lead to health benefits, but they are limited by other adverse effects. The studies of Allegre et al. [28] and Zekioglu et al. [29] underline the practice of more intense training in alexithymic athletes, which could be the origin of burnout. Alexithymic athletes fall into an over-adapted state without reflecting on their condition or internal experience, which can induce mental fatigue [26]. This is probably what subjects practicing sport for more than 5 h per week experience.

In our study, alexithymics were slightly younger than borderline or non-alexithymic participants. According to Mattila et al., [47] the prevalence of alexithymia grew steadily with increasing age. This is consistent with the emotional changes associated with later life [48]. However, we see that it is the opposite in our study. This difference is probably due to their lack of experience about student life stressors. Younger students are probably more stressed about “students life” because they are still discovering things (exams, leaving home, etc.) and engage in more sport in order to have positive feelings and reduce pressure/sensation seeking.

This is an original investigation into the relationship between alexithymia and sport in a population of students. Student recruitment was consecutive, with no selection bias, and we used a well-established questionnaire for the assessment of alexithymia. This study provides additional information, such as the need for psychological support, to help improve pastoral care for students.

However, this study has some limitations. First, we intended to assess the occurrence of alexithymia and not its prevalence in our small sample size (253), as this could enhance the influence of the chance factor, and the inference of the whole study would be questionable. Furthermore, our sample of athletic people was relatively small, at only 85 subjects among the 253 participants included. The proportion of students engaged in sport (33.6%) was lower than in the French population overall (43% practicing more than once per week). This is certainly explained by the more sedentary student lifestyle and the financial precariousness of graduates. Indeed, most students work outside class to make ends meet. Female over-representation in this study population might also be a partial additional explanation. Another limitation was the subjective self-assessment of athletic status, as it was based on each participant’s interpretation of the question “Do you regularly practice some sport?” The lack of a clear definition of sport practice that could be considered athletic may have introduced a bias. Indeed, answers could have vacillated between a real and an ideal perception of each participant’s own practice. To overcome this uncertainty, the question specifying time devoted to weekly sports practice (<5 h vs. >5 h) therefore seems more relevant. Whether sports training contributes to the development of alexithymic traits remains a compelling question, since alexithymia itself might be considered an element of performance in high-risk sports. The prospect of training at a basic level or of a certain type to enhance alexithymia and thus, to influence performance, is of crucial importance to both athletes and coaches.

## 5. Conclusions

Our results demonstrated the high occurrence of alexithymia in French university students and the possible relationship between alexithymia and the amount of athletic training practice. Given its well-recognized health benefits, physical exercise is widely promoted. However, it is also easy for physical activity to become excessive, and the adverse psychological consequences of this should be highlighted. Our work confirms the crucial need for psychological support for students. Further studies are needed to clarify the characteristics of a specific association between alexithymia, amount of training, type of sport, and the level of practice.

## Figures and Tables

**Table 1 healthcare-10-00788-t001:** Descriptive characteristics of student participants according to sport practice.

	Total (*n* = 253)	Athletic (*n* = 85)	Non-Athletic (*n* = 168)	*p*-Value
**Sex (women)**	194 (76.7)	58 (68.2)	136 (80.9)	0.019
**Age (mean (SD))**	20.6 (3.0)	20.9 (4.4)	20.4 (2.0)	0.311
**Type of sport**				
Nothing	168 (66.4)	-	168 (100.0)	-
Martial arts	10 (4.0)	10 (11.8)	-	-
Team sports	9 (3.6)	9 (10.6)	-	-
Dance	9 (3.6)	9 (10.6)	-	-
Racket sport	6 (2.4)	6 (7.1)	-	-
Endurance sport	24 (9.5)	24 (28.2)	-	-
Bodybuilding/fitness	18 (7.1)	18 (21.2)	-	-
Sport relaxation	2 (0.8)	2 (2.4)	-	-
Track and field	2 (0.8)	2 (2.4)	-	-
Individual sport	5 (2.0)	5 (5.9)	-	-
**Amount of sport (per week)**				
Nothing	168 (66.4)	-	168 (100.0)	-
<5 h	46 (18.2)	46 (54.1)	-	-
>5 h	39 (15.4)	39 (45.9)	-	-
**Level of practice**				
Recreational	67 (26.5)	67 (78.8)	-	-
Competition	18 (7.1)	18 (21.2)	-	-

We used the mean (±SD) to present continuous variables or as *n* (%) for categorical variables.

**Table 2 healthcare-10-00788-t002:** Alexithymia and sport practice characteristics.

	Non-Alexithymia (*n* = 85)	Borderline (*n* = 92)	Alexithymia (*n* = 76)	Total	*p*-Value
**Age**	21.1 (2.4)	20.7 (4.2)	19.9 (1.3)	20.6 (3.0)	<0.001
**Sex (women)**					
Men	26 (30.6)	22 (23.9)	11 (14.5)	59	0.054
Women	59 (69.4)	70 (76.1)	65 (85.5)	194	
**Amount of sport (per week)**					
Nothing	58 (68.2)	57 (62.0)	53 (69.7)	168	0.030
<5 h	21 (24.7)	17 (18.5)	8 (10.5)	46	
>5 h	6 (7.1)	18 (19.6)	15 (19.7)	39	
**Level of practice**					
Nothing	58 (68.2)	57 (62.0)	53 (69.7)	168	0.812
Spare time	21 (24.7)	27 (29.3)	19 (25.0)	67	
Competition	6 (7.1)	8 (8.7)	4 (5.3)	18	

The mean (±SD) was used to present age and as *n* (%) for other variables.

## Data Availability

Not applicable.

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
