# Peer review of "Occurrence of Alexithymia and Its Association with Sports Practice from a Sample of University Students: Results from a French Cross-Sectional Study"

_healthcare, 2022, doi:10.3390/healthcare10050788_

Round 1
Reviewer 1 Report
The study sought to examine the association between alexithymia and participation in sports among University students in a cross-sectional design.
My major concern is that the authors have not been able to interpret their data to shed light on how a physical activity or exercise participation may be associated with alexithymia. Or do the authors suggest that exercise participation is a coping mechanism or strategy against alexithymia? Several other questions arise, for example, do people exercise because they may be experiencing alexithymia, or does the latter influence exercise participation? Given the well-recognized health benefits of physical exercise to human health, how do the authors place their study findings in the context of the physical activity recommendations? Also, what proportion of the physically fit population is alexithymic? Generally, the authors failed to provide detailed scientific background information to the paper, and hence, leaves many authors with several questions.
It would be helpful for the authors to provide detailed, relevant literature in the introduction and in the Discussion to improve clarity while disentangling any association (if any) between physical activity and alexithymic which is different from the usual participation in physical activity.
Similarly, the methods sections are very scanty and do not adequately describe the study. For instance, the eligibility criteria were not well and fully described. The authors should endeavor to improve the quality of the scientific basis of the study.
Reviewer 2 Report
General comments
This research as presented in this article was scientifically well conceptualized and executed. However, there a number of areas that require attention to bring it to a level of publication. The authors should attend to the following:
Introduction
This section is well designed and well-written.
Materials and methods
This section is well designed and well-written.
Results section
This section is well designed and well-written.
Discussion section
The authors should add the relevant references about the Alexithymia and sports, these studies will allow improving the manuscript.
The authors should add these articles
Jodat, H., Ghasempour, A., & Tavakoli, A. (2015). Comparison of Alexithymia in athlete and non-athlete students. Annals of Applied Sport Science, 3(3), 17-24.
Barlow, M., Woodman, T., Chapman, C., Milton, M., Stone, D., Dodds, T., & Allen, B. (2015). Who takes risks in high-risk sport?: The role of alexithymia. Journal of Sport and Exercise Psychology, 37(1), 83-96.
Woodman, T., & Welch, C. (2021). Alexithymia and the Anxiolytic Effect of Endurance Running. The Sport Psychologist, 1(aop), 1-7.
Iacolino, C., Pellerone, M., Formica, I., Concetta Lombardo, E. M., & Tolini, G. (2017). Alexithymia, body perception and dismorphism: a study conducted on sportive and non-sportive subjects. Clinical Neuropsychiatry, (6).
Proença Lopes, C., Allado, E., Poussel, M., Hamroun, A., Essadek, A., Albuisson, E., & Chenuel, B. (2022). An Association between Alexithymia and the Characteristics of Sport Practice: A Multicenter, Cross-Sectional Study. In Healthcare (Vol. 10, No. 3, p. 432). MDPI.
Allegre, B., Souville, M., Noel-Jorand, M. C., Pellegrin, L., & Therme, P. (2007). Intensive Physical activity and alexithymia: results from swimmers' discourse analysis. Psychological reports, 100(3_suppl), 1129-1139.
Reviewer 3 Report
This manuscript has a huge lack of theoretical and methodological content that is necessary for its publication, so I cannot recommend its publication in this state. The main deficiencies are presented below:
- In the 42-43 lines, the statement "Numerous studies have evaluated..." leads to only one reference, which is paradoxical, since numerous studies only lead to one reference.
- The same issue happens in 51-52 with the statement "Sport has been shown to be strongly associated with alexithymia..." which leads only to another single reference.
- In general, the introduction to the manuscript is extremely short and limited. The authors should expand the relationship between sport and alexithymia with the description of the numerous articles that have described this relationship over time and justify the research.
- The psychometric properties, validity and reliability of the instruments used in the research are lacking.
- There are hyphens between some words throughout the text. It should be thoroughly reviewed.
- In the Discussion there are referenced researches that should have been presented firstly in the Introduction to justify the research carried out.
- There is no comparison with other studies dealing with alexithymia and sport (There are a lot of them in the recent years).Remember to include them in the research justification in the introduction section.
- The significant relationship between alexithymia and age is not analysed in Discussion.
- Conclusions are extremely short.
- Most of the articles in the bibliography are from more than five years ago.
Round 2
Reviewer 1 Report
The current manuscript shows an improvement over the previous version. The manuscript may be accepted after English editing.
Thank you.
Reviewer 3 Report
Congratulations to the authors for a considerable improvement. However, there are still some issues to improve. Still to fix:
- The significant relationship between alexithymia and age is not analysed in Discussion.
- There is no comparison with other studies dealing with alexithymia and sport (There are a lot of them in the recent years): In this case, the improvement made is scarce and the results of this study need to be expanded and compared with more recent similar studies.
